# An Integrated Approach for Treatment of Acute Type A Aortic Dissection

**DOI:** 10.3390/medicina57111155

**Published:** 2021-10-24

**Authors:** Igor Vendramin, Andrea Lechiancole, Daniela Piani, Sandro Sponga, Concetta Di Nora, Daniele Muser, Uberto Bortolotti, Ugolino Livi

**Affiliations:** 1Cardiothoracic Department, Division of Cardiac Surgery, University Hospital of Udine, 33100 Udine, Italy; andrea.lechiancole@asufc.sanita.fvg.it (A.L.); daniela.piani@asufc.sanita.fvg.it (D.P.); sandro.sponga@asufc.sanita.fvg.it (S.S.); concetta.dinora@asufc.sanita.fvg.it (C.D.N.); uberto48@gmail.com (U.B.); ugo.livi@asufc.sanita.fvg.it (U.L.); 2Cardiothoracic Department, Division of Cardiology, University Hospital of Udine, 33100 Udine, Italy; daniele.muser@asufc.sanita.fvg.it; 3Department of Medical Area (DAME), Division of Cardiac Surgery, University of Udine, 33100 Udine, Italy

**Keywords:** acute dissection, arch replacement, FET

## Abstract

*Background and objective:* We reviewed a single-institution experience to verify the impact of surgery during different time intervals on early and late results in the treatment of patients with type A acute aortic dissection (A-AAD). *Materials and Methods*: From 2004 to 2021, a total of 258 patients underwent repair of A-AAD; patients were equally distributed among three periods: 2004–2010 (Era 1, *n* = 90), 2011–2016 (Era 2, *n* = 87), and 2017–2021 (Era 3, *n* = 81). The primary end-point was to assess whether through the years changes in indications, surgical strategies and techniques and increasing experience have influenced early and late outcomes of A-AAD repair. *Results*: Axillary artery cannulation was almost routinely used in Eras 2 (86%) and 3 (91%) while one femoral artery was mainly cannulated in Era 1 (91%) (*p* < 0.01). Retrograde cerebral perfusion was predominantly used in Era 1 (60%) while antegrade cerebral perfusion was preferred in Eras 2 (94%,) and 3 (100%); (*p* < 0.01). There was a significant increase of arch replacement procedures from Era 1 (11%) to Eras 2 (33%) and 3 (48%) (*p* < 0.01). A frozen elephant trunk was mainly performed in Era 3. Hospital mortality was 13% in Era 1, 11% in Era 2, and 4% in Era 3 (*p* = 0.07). Actuarial survival at 3 years is 74%, in Era 1, 78% in Era 2, and 89% in Era 3 (*p* = 0.05). *Conclusions*: With increasing experience and a more aggressive approach, including total arch replacement, repair of A-AAD can be performed with low operative mortality in many patients. Patient care and treatment by a specific team organization allows a faster diagnosis and referral for surgery allowing to further improve early and late outcomes.

## 1. Introduction

Type A acute aortic dissection (A-AAD) has long been recognized as an ominous and highly lethal form of aortic disease. Early pathological studies have shown that death occurred within 24 h in almost 60% of patients diagnosed with A-AAD, supporting the need for timely management [1]. However, despite early treatment, initial results of repair of A-AAD were plagued by an operative mortality as high as 58% [2,3,4].

Following the first successful repair of A-AAD, with excision and graft replacement of the ascending aorta, reported by DeBakey and his team in Houston, Texas [5], improvements in preoperative diagnosis, intraoperative management and postoperative care have led to a substantial improvement of surgical results in most centers [6]. In the past decades there have been considerable changes in the indications and surgical strategies for patients presenting with A-AAD and these are mainly represented by a more aggressive approach which includes extension of the repair particularly to the aortic arch and root [7,8].

Even if saving patient life is certainly the first and foremost goal of A-AAD repair, long-term survival, reoperation-free survival, and quality of life, cannot be considered of secondary importance anymore. As such, an extended repair appears often more reasonable as initial strategy compared to a limited, albeit lifesaving, treatment. Notwithstanding, the debate on the best intraoperative strategy and whether surgical and technological improvements have contributed to provide a superior patient outlook is still open.

In the present manuscript, we reviewed a single-institution experience to verify the impact of surgical strategy on patient outcomes during different time intervals and whether the development of a specialized aortic team has contributed to improve early and long-term results in the treatment of A-AAD.

## 2. Methods

During a 17-year interval, from 2004 to 2021, a total of 258 patients underwent repair of A-AAD at tour Institution. Patients were equally distributed among the three periods: 2004–2010 (Era 1, *n* = 90), 2011–2016 (Era 2, *n* = 87), and 2017–2021 (Era 3, *n* = 81). The identification of the 3 different time intervals was mainly based on differences in the surgical technique and overall patient management such as the use of selective antegrade cerebral perfusion, right axillary artery cannulation, and development of a specialized aortic team with faster patient referral from other regional centers. The aim of this study was to verify, as primary endpoint, whether, through the years, changes in indications, surgical strategies and techniques and increasing experience have influenced early and late outcomes of such patients. The study was approved by the local ethical committee and patient informed consent was waived owing to the retrospective nature of the analysis.

*Preoperative clinical data:* Table 1 summarizes the most pertinent preoperative characteristics which were substantially similar in the 3 groups. In particular, mean age was 64 ± 12 years in Era 1, 63 ± 12 years in Era 2 and 66 ± 13 years in Era 3 (*p* = 0.13) and most patients were males. Moreover, over 70% of patients had arterial hypertension and 5% had undergone a previous cardiac operation.

Clinical presentation was characterized by cardiac tamponade with shock in >30%, neurological damage in 16% and splanchnic malperfusion in 10% of cases, without significant differences among the three surgical eras.

*Surgical indications and techniques*: During the study period, the surgical strategies remained substantially stable. Patients presenting with signs and symptoms of A-AAD were always treated on an emergency basis as soon as the diagnosis was confirmed by transthoracic 2D echo and/or angio-computed tomography (CT). Potential contraindications were progressively reduced, current indications including patients with systemic malperfusion, unless with irreversible neurological damage, and advanced age.

The surgical strategy was mainly dictated by the preoperative imaging and the intraoperative findings. Graft replacement of the ascending aorta, always extended to include the hemiarch, was considered as a limited repair, performed when the entry tear was found only in the ascending aorta; dilatation of the aortic arch or presence of arch tears, evidenced by routinely performing an open distal anastomosis, represented an indication to replace both the ascending aorta and arch. Management of the aortic root depended upon its size, morphology and function, and involvement of the aortic valve [8].

All operations were performed through a standard median sternotomy. Cannulation for cardiopulmonary bypass (CPB) was initially through the femoral vessels while in most recent years the right axillary artery was routinely utilized. Myocardial protection was always achieved with St. Thomas 1 crystalloid cardioplegia. Deep hypothermic circulatory arrest with retrograde cerebral perfusion were initially favored but then replaced by moderate hypothermia and continuous antegrade cerebral perfusion using the right axillary artery and direct cannulation of the left carotid artery. When replacing the arch, the distal aortic stump was sandwiched between Teflon strips and covered with biological glue; arch reconstruction was accomplished mainly by the classic or frozen elephant trunk (ET) procedure using a quadrifurcated graft. The distal suture was carried out while a tip-cut Foley catheter, inflated into the graft, provided splancnic perfusion, which was then replaced by cannulating the lateral branch of the graft. Anastomosis of the epiaortic vessels and the proximal aortic suture line, as well as any other procedure on the aortic root or valve, were performed during rewarming.

Patients having aortic valve or root replacement with a mechanical prosthesis were routinely kept on life-long oral anticoagulants; in all the others, antiplatelet medications were administered unless specific thrombotic risk factors were present.

*Patient follow-up:* Following repair of A-AAD, patient follow-up included clinical and 2D echo re-evaluation at one and six months after discharge and yearly thereafter. Since 2010, CT were performed annually for the first 5 years and then whenever considered indicated. All data obtained were entered in a specific database collecting information on clinical status, incidence and causes of postoperative complication, and results of imaging investigations.

*Statistical analysis:* Continuous variables were expressed as mean ± standard deviation if normally distributed or median (25th–75th percentile) if not normally distributed. All continuous variables were tested for normal distribution using the 1-sample Kolmogorov–Smirnov test. Categorical data were expressed as counts and percentages. Differences in continuous variables between more than two groups were assessed using the one-way ANOVA test or Kruskall–Wallis test, where appropriate. Chi-square or Fisher’s exact test, where appropriate, were computed to assess differences in categorical variables.

A univariable and multivariable log-binomial regression analysis was performed to evaluate the relative risk (RR) of baseline covariates associated with 30-day mortality. All potential confounders were initially entered into the multivariable model on the basis of known clinical relevance or imbalances between groups identified in univariable analyses; then, a model reduction was performed by excluding variables that did not have any confounding effect (<10% variation in the RR). Adjusted Cox proportional hazards analysis (including clinically relevant confounders: age, gender, LVEF, hypertension, diabetes, chronic kidney disease, coronary artery disease, and history of atrial fibrillation) was used to test the association between the Era of surgery and long-term outcome events. Survival curves were generated by the Kaplan–Meier method and compared by the Log-Rank test. Two-tailed tests were considered statistically significant at the 0.05 level. All the analyses were performed using IBM SPSS version 24.0 software (SPSS Inc., Chicago, IL, USA).

## 3. Results

*Surgical data*: Axillary artery cannulation for CPB was almost routinely used in Era 2 (75 patients, 86%) and Era 3 (74 patients, 91%) compared to Era 1 (9 patients, 9%) (*p* < 0.01) while one femoral artery was mainly cannulated in Era 1 (81 patients, 91%) (*p* < 0.01).

Retrograde cerebral perfusion for neurologic protection was predominantly used in Era 1 (54 patients, 60%), in only 5 patients (6%) of Era 2 and never in Era 3 (<0.01). Instead, antegrade selective perfusion was the preferred method of cerebral protection in Eras 2 (82 patients, 94%) and 3 (81 patients, 100%) and only in 25 patients (28%) of Era 1 (*p* < 0.01).

There was a significant increase, through the years, of arch replacement procedures which were performed in 10 patients (11%) in Era 1, 29 (33%) in Era 2 and in 39 (48%) in Era 3 (*p* < 0.01). A classic ET was applied in 10 patients of Era 1 (11), in 28 (32%) of Era 2, and in 26 (32%) of Era 3. A frozen ET was performed in 1 patient (1%) of Era 2 and in 13 patients (16%) of Era 3, but in no patients of Era 1. Other associated surgical procedures on the aortic root or aortic valve are shown in Table 2.

Mean duration of CPB was 224 ± 67, 231 ± 86, and 206 ± 73 min in Eras 1, 2, and 3, respectively (*p* = 0.14); mean aortic cross-clamp (ACC) time was 118 ± 51 min in Era 1, 122 ± 58 min in Era 2, and 119 ± 53 min in Era 3 (*p* = 0.84); finally, mean splancnic arrest time was 50 ± 23, 48 ± 22, and 39 ± 15 min in Eras 1, 2, and 3, respectively (*p* < 0.01).

*Early mortality and morbidity*: There were 12 hospital deaths in Era 1 (13%), 10 in Era 2 (11%), and 3 in Era 3 patients (4%) (*p* = 0.07) and *p* = 0.03 Era 1 vs. Era 3, as reported in Table 3. Causes of death were similar among the three groups being mainly represented by hemorrhagic shock (4%) and multiorgan failure (2%).

As shown in Table 4, clinical presentation with tamponade/shock or neurological damage and circulatory arrest time were independently associated with 30-day mortality.

Most relevant postoperative complications include acute renal failure in 38 patients of Era 1 (42%), 34 of Era 2 (39%), and 41 of Era 3 (51) (*p* = 0.3) with the need for renal replacement therapy in 17%, 18%, and 15% of cases, respectively. Chest re-exploration for bleeding was required in 17 patients of Era 1 (19%), 19 (22%) of Era 2, and 5 (6%) of Era 3 (*p* = 0.02). In Era 3, a lower percentage of patients required high inotropic support (21%) compared to Era 1 (38%) and Era 2 (34%), *p* = 0.04. No significant differences were reported among neurological complications, although lower rates of postoperative coma and permanent neurological deficit were noted in Era 3.

Transient neurological deficits occurred in 12 (13%), 12 (14%), and 11 patients (14%) of Eras 1, 2, and 3, respectively (*p* = 0.9), while postoperative paraplegia was observed in <2% of cases in the 3 groups.

Median intensive care stay was 8 days (range, 1–26 days) in Era 1, 8 days (range, 1–24 days) in Era 2, and 7 days in Era 3 (range, 1–17 days) (*p* = 0.25), while median hospital stay was 20 (range, 5–65 days), 22 (range, 4–53 days), and 18 days (range, 4–49 days) in Eras 1, 2, and 3, respectively (*p* = 0.17).

*Late results and survival*: There were 64 late deaths, 41 (45%) in patients operated during Era 1, 18 (21%) in Era 2, and 5 (6%) in Era 3 (Table 4). Of these, 20 were aortic-related: 11 (27%) in Era 1, 6 (33%) in Era 2, and 3 (60%) in Era 3 (Table 5). Actuarial survival at 1 year is 89% for Era 1, 84% for Era 2, and 89% for Era 3 patients; at 3 years is 73% for Era 1, 78% for Era 2, and 88% for Era 3 patients; at 8 years is 51% for Era 1 and 62%% for Era 2 patients (Figure 1).

During the follow-up reintervention on the thoracic aorta was required in 10 patients of Era 1 (11%), 12 of Era 2 (14%), 2 of Era 3 (2.5%). Type of reoperation and median time from index operation are indicated in Table 6.

Actuarial freedom from reoperation at 1 year is 96% for Era 1, 88% for Era 2, and 99% for Era 3 patients; at 3 years is 92% for Era 1, 81% for Era 2, and 96% for Era 3 patients; at 8 years is 85% for Era 1 (Figure 2).

After adjustment for clinically relevant confounders, Era 3 patients had a significantly better long-term survival with a 68% lower risk of death compared to Era 1 patients (adjusted HR 0.32, 95% CI 0.14–0.72; *p* = 0.006) while no significant differences were observed for Era 2 patients (adjusted HR 0.80, 95% CI 0.47–1.35; *p* = 0.39). Compared to Era 1 patients, no significant differences in the long-term risk of reintervention were observed among Era 2 patients (adjusted HR 1.54, 95% CI 0.63–3.75; *p* = 0.33) nor Era 3 patients (adjusted HR 0.41, 95% CI 0.08–2.00; *p* = 0.27).

*Classic* vs. *frozen ET*: All patients having aortic arch replacement with a frozen ET procedure were operated in Era 3. Of these, 8 (61%) were males compared to 58% (15 patients) having a classic ET during the same Era (*p* = 0.89); frozen ET patients were younger (mean age 55 ± 9 years) then those with a classic ET (mean age 66 ± 2 years) (*p* = 0.07). In patients with a frozen ET mean CPB, 204 ± 49 vs. 265 ± 84 min (*p* = 0.02) and ACC times, 140 ± 49 vs. 151 ± 61 min (*p* = 0.58) were shorter as well as the mean circulatory arrest time (29 ± 7 vs. 50 ± 14 min, *p* < 0.01). Median ICU stay was 5 days (1–24 days) for frozen ET vs. 7 days (1–28 days) for classic ET patients (*p* = 0.55), while median hospital stay was 15 days (1–53 days) vs. 21 days (1–65 days) (*p* = 0.32) (Table 7).

Patients who underwent classic ET were more likely to suffer from acute renal failure (72% vs. 38%, *p* = 0.07) and to require renal replacement treatment (30% vs. 0%, *p* = 0.03).

Hospital mortality was 0% in patients with a frozen ET in all eras and 10%, 11% and 4% in Eras 1, 2, and 3, respectively, in patients with classic ET (Figure 3).

There were two late deaths in patients with a classic ET, one of these was aortic-related and one due to infection. One patient in the classic ET group was reoperated (AVR). There were no late deaths or reoperations in the frozen ET group. Actuarial survival at 2 years is 87 ± 7% for classic ET and 100% for frozen ET patients (*p* = 0.2).

## 4. Discussion

Almost 70 years have elapsed after the first report of a successful operation in a patient with A-AAD [5]. Since then, enormous improvements have occurred in A-AAD treatment in all fields, including surgical techniques, intraoperative myocardial and cerebral protection, and synthetic graft technology, all of which have resulted in progressively better early and late mortality. When reviewing the early historical reports one becomes aware of the influence that initial efforts to repair A-AAD have had on the evolution of surgical strategies.

As described by DeBakey, who provided the well-known and still used classification, type I A-AAD repair initially consisted ‘*essentially in transection of the ascending aorta with the use of temporary cardiopulmonary bypass and obliteration of the false lumen by approximation of the inner and outer walls of the dissecting process by means of a continuous suture both proximally and distally, followed by end-to-end anastomosis of the transected aorta*’ [2]. Instead, in Type II A-AAD ‘*surgical treatment for these aneurysms consists essentially in resection and graft replacement of the entire ascending aorta using temporary cardiopulmonary bypass’* [2]. With growing experience, graft replacement of the ascending aorta was subsequently applied to all patients with A-AAD including repair or replacement of the aortic root [9].

Replacement of the ascending aorta, possibly including the hemiarch as routinely performed in our center, is certainly the simplest method to deal with A-AAD, especially if the aortic arch is tear-free and not dilated, providing excellent results in most patients with A-AAD [10]. However, it is currently evident that Occam’s razor rule of ‘*the simplest solution is always the better*’ [11] may not be applicable in this specific setting when the long-term outcomes after A-AAD repair become a major objective; in fact, fragility of the residual dissected tissue coupled with possible progressive false lumen enlargement render survivors of A-AAD repair prone to develop complications, such as pseudoaneurysm formation, arch dilatation or false lumen expansion, which require late, higher-risk reoperations [12,13]. Nevertheless, which should be the best surgical strategy in A-AAD repair remains at present still debated, since the choice of operation may also depend on causes not strictly related to the underlying pre and intraoperative findings such as patient clinical and neurologic conditions, age and comorbidities, and experience of the caring surgeon; all these factors may at times suggest to avoid a complex, albeit more radical, operation in favor of a limited, low-risk repair.

A special consideration should be also given to the evolution of surgical techniques in treating the aortic arch in A-AAD. Initially, graft replacement of the arch was associated with reimplanting the epiaortic vessels contained into an island of arch tissue [14]; the straight aortic graft was then replaced by individual reattachment of the arch vessel using a tri- or quadrifurcated graft [15]. Metallic stents have been also used to stabilize the distal false lumen but despite initial success, these devices have been abandoned because of many postoperative complications that have been observed related to the specific stent design [16,17]. Recently, a more stable arch repair has been obtained with the introduction of the frozen ET, which at variance with the procedure originally described by Borst et al. almost 40 years ago, but using then same principle [18], has provided the means for effective arch replacement. Using a hybrid graft with a distal expandable stent, a more rigid landing zone is offered, should distal late complication ensue requiring future reoperations or endovascular treatments [18,19,20]. Furthermore, gratifying results have been obtained also with bare stents during repair of aortic arch dissection using a trans-femoral approach [21,22].

From our analysis some important considerations can be drawn, to provide further data on this controversial issue. Hospital mortality has significantly dropped from 13% in Era 1 to 4% in Era 3; this result, with improvement of an already acceptable mortality, is particularly gratifying when considering that in the last interval there has been a significant increase of aortic arch operations while the patient profile remained substantially unchanged. Through the years, there have been some important technical modifications such as a progressively more extended use of the EF technique, with routine employment in the recent years of the frozen ET, and of axillary artery cannulation for CPB. The use of the ET follows our trend for a more aggressive approach in patients with A-AAD, switching from the traditional ascending aorta and hemiarch, mainly favored in Era 1, to include subsequently total aortic arch replacement in most A-AAD repairs. In our opinion, availability of the frozen ET technology has facilitated aortic arch replacement contributing to improve the operative results since it can be used also in quite complex settings for hybrid treatments [23]; moreover, repair appears more stable since none of the patients required reoperation during the follow-up while in some cases thoracic endovascular repair successfully treated late aortic-related complications. These data support the results obtained by others who report an operative mortality <7%, 74% late survival, and 84% freedom from reoperation at 8 years using the frozen ET [24]. Despite our still limited experience, the effectiveness of the frozen ET is also supported by our results comparing patients having a classic vs. a frozen ET procedure. Frozen ET, which is our preferred technique for aortic arch replacement in the current era, has provided evident advantages from the surgical viewpoint; indeed, CPB and ACC times have been reduced as well as duration of overall splanchnic ischemic time. Furthermore, it appears to give more stability to the repair of A-AAD since no reoperations due to aortic-related complications have occurred during the follow-up. These results are also supported by the multivariable analysis indicating the reduced AAC time in patients operated with the frozen ET technique as positively influencing early mortality.

Cannulation of the right axillary artery has become our preferred method for perfusion, almost eliminating the use of one femoral artery which, providing a retrograde flow, may predispose to cerebral embolization with permanent neurologic deficits [25]. Axillary artery cannulation, in our experience, by maintaining a forward flow during the open distal anastomosis, facilitates also antegrade cerebral perfusion which is then completed only by selective cannulation of the left carotid artery; this allows to avoid more cumbersome maneuvers minimizing the risk of iatrogenic injury to the epiaortic vessels.

Another interesting result is represented by the significant reduction of the splanchnic ischemic time, considered as the time employed, after release of the aortic cross-clamp, to inspect the aortic arch, resect the aortic tissue, detach the epiaortic vessels and prepare the distal aortic stump. By adopting the use of a catheter through the graft, distal perfusion can be resumed earlier reducing the splanchnic ischemic damage; indeed, we observed few cases of paraplegia while there was no substantial difference among the three groups in the incidence of postoperative acute kidney injury. These results may depend on the fact that the number of patients, comparable in the three groups, may still be too small to verify whether changes in surgical strategies have provided real benefits besides a reduction of operative mortality; therefore, larger patient populations might be required to confirm a positive trend and our favorable impression.

An explanation for the progressive improvement of the surgical results may be found considering the evolution which took place in our center in the management of patients with A-AAD. Initially, few dedicated surgeons were assigned to this complex task which included also training of other staff members. Furthermore, an aortic team was created including cardiac and vascular surgeons, cardiologists, anesthesiologists, and interventional radiologists to evaluate each patient but also to establish definite protocols for diagnosis and treatment of A-AAD patients; then, a specific network was created with other regional centers, an organization frequently advocated [5], to share such protocols and to assure a fast diagnosis and referral to the surgical facilities with reduction of the interval from diagnosis to treatment. Indeed, in Era 3 patients there has been a reduced incidence of shock due to cardiac tamponade and cerebral or systemic malperfusion. As a consequence of training and spreading of surgical knowledge, all senior surgeons in our center currently concur on the care of patients presenting with A-AAD. We consider the effectiveness of this organization as the main determinant of the steady improvement of our surgical results, despite an increasing number of complex procedures and considering that what we have reported is a multi-surgeon experience.

This study is an analysis of a single-center experience in managing A-AAD in different surgical eras. The main limitation is represented by its retrospective, observational nature; nevertheless, we believe that the information provided, by comparing three groups of patients treated in different time intervals may be useful to support specific surgical strategies. Indeed, by evaluating the impact of time on early and late outcomes of patients undergoing A-AAD repair, we observed that modifications of some intraoperative procedures, technological advancements and better organization of patient referral introduced in our practice, have undoubtedly contributed to improve the outlook of patients with A-AAD.

In conclusion, A-AAD remains a challenging pathology which requires timely diagnosis and appropriate repair. Our results demonstrate that by increasing experience, a more aggressive approach with total arch replacement and patient care by a specific organization allows to improve early and late outcomes in most patients presenting with A-AAD.

## Figures and Tables

**Figure 1 medicina-57-01155-f001:**
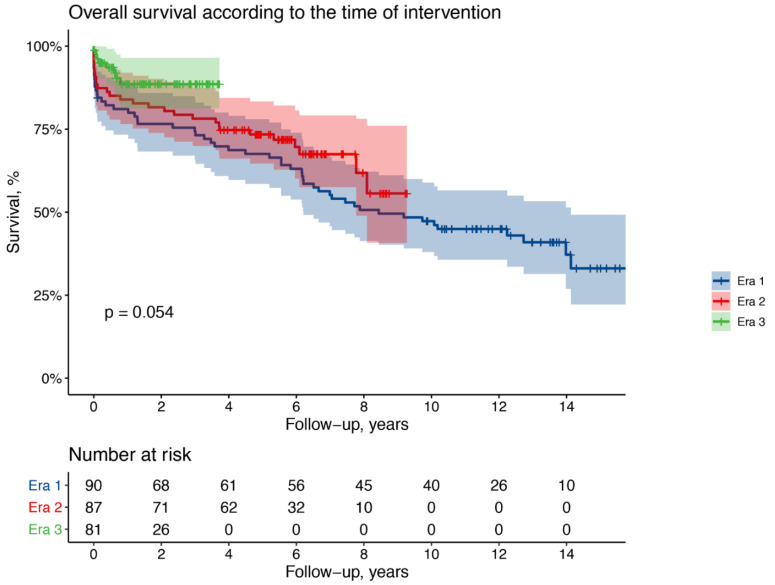
Kaplan–Meier curves showing overall survival according to the era of intervention.

**Figure 2 medicina-57-01155-f002:**
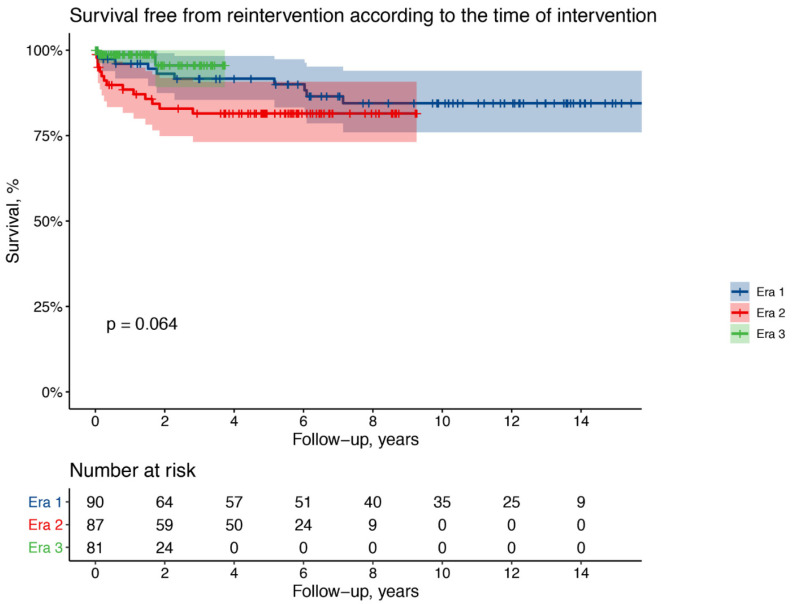
Kaplan–Meier curves showing the freedom from reoperation according to the era of the first intervention.

**Figure 3 medicina-57-01155-f003:**
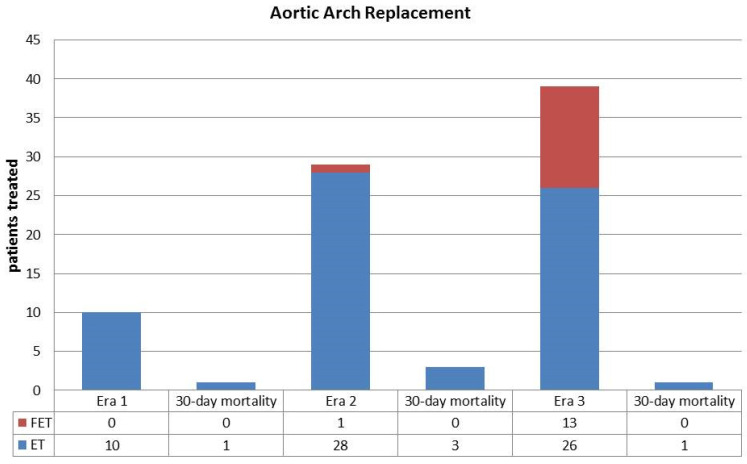
Comparison of number of classic elephant trunk (ET) and frozen (FET) procedures and related operative mortality in the three different surgical eras.

**Table 1 medicina-57-01155-t001:** Baseline characteristics.

	Era 1 (*n* = 90)	Era 2 (*n* = 87)	Era 3 (*n* = 81)	*p* Value
*Clinical profile*				
Male sex, *n* (%)	58 (64)	67 (77)	41 (51)	0.02
Age (years, mean ± sd)	64 ± 12	63 ± 12	66 ± 13	0.13
LVEF, %	56 ± 8	51 ± 8	56 ± 9	<0.01
	*n. (%)*	*n. (%)*	*n. (%)*	
Chronic renal failure	22 (24)	17 (19)	12 (15)	0.28
COPD	11(12)	5 (6)	4 (5)	0.14
Previous cardiac surgery	4 (4)	6 (7)	2 (2)	0.39
Bicuspid aortic valve	6 (7)	4 (5)	2 (2)	0.43
CAD	7 (8)	8 (9)	4 (5)	0.56
Previous stroke	4 (4)	2 (2)	0 (0)	0.15
Chronic AF	9 (10)	7 (8)	12 (15)	0.35
*Presentation*				
Tamponade/shock/hypotension	31 (34)	33 (38)	18 (22)	0.07
Syncope	15 (17)	21 (22)	8 (11)	0.39
Focal neurologic damage	13 (14)	16 (18)	13 (16)	0.77
Coma	0 (0)	2 (2)	2 (3)	0.17
Splanchnic malperfusion	14 (16)	10 (11)	3 (4)	0.01
- *Mesenteric*	*3 (3)*	*1 (1)*	*0 (0)*	
- *Renal*	*6 (7)*	*6 (7)*	*2 (2)*	
- *Lower limb*	*5 (6)*	*4 (5)*	*2 (2)*	
*Risk factors*				
Arterial hypertension	64 (71)	69 (79)	63 (78)	0.36
Smoking	23 (26)	22 (25)	29 (36)	0.23
Dyslipidemia	8 (9)	10 (11)	18 (22)	0.03
Diabetes	4 (4)	3 (3)	6 (7)	0.47

COPD = chronic obstructive pulmonary disease; CAD = coronary artery disease; AF = atrial fibrillation.

**Table 2 medicina-57-01155-t002:** Surgical data.

	Era 1 (*n* = 90)	Era 2 (*n* = 87)	Era 3 (*n* = 81)	*p* Value
*Surgical procedures*	*n* (%)	*n* (%)	*n* (%)	
Aortic Arch Replacement	10 (11)	29 (33)	39 (48)	<0.01
- Elephant Trunk	10 (11)	28 (32)	26 (32)	<0.01
- Frozen Elephant Trunk	0 (0)	1 (1)	13 (16)	<0.01
Aortic Root Replacement	11(12)	14 (15)	7 (8)	0.16
- Bentall	9 (10)	12 (14)	4 (5)	
- Reimplantation	2 (2)	2 (2)	3 (4)	
- Remodeling	0 (0)	2 (2)	0 (0)	
Aortic Valve Replacement	8 (9)	1 (1)	3 (4)	0.04
- Mechanical	5 (6)	0 (0)	0 (0)	
- Biological	3 (3)	1 (1)	3 (4)	
Aortic Valve Repair	5 (6)	7 (8)	3 (4)	0.62
Associated CABG	3 (3)	6 (7)	6 (7)	0.45
Associated MV surgery	1 (1)	0 (0)	0 (0)	0.89
Mean CPB time, minutes ± SD	224 ± 67	231 ± 86	206 ± 73	0.14
Mean ACC time, minutes ± SD	118 ± 51	122 ± 58	119 ± 53	0.84
Mean circulatory arrest, minutes ± SD	50 ± 23	48 ± 22	39 ± 15	<0.01
*Operative Setting*	*n (%)*	*n (%)*	*n (%)*	
Axillary artery cannulation	8 (9)	75 (86)	74 (91)	<0.01
Femoral artery cannulation	81 (91)	12 (14)	7 (9)	<0.01
Antegrade cerebral perfusion	25 (28)	82 (94)	81 (100	<0.01
Retrograde cerebral perfusion	54 (60)	5 (6)	0 (0)	<0.01

AA = ascending aorta; AV = aortic valve; CABG = coronary artery bypass grafting; MV = mitral valve; ET = elephant trunk; CPB = cardiopulmonary bypass; ACC = aortic cross-clamp.

**Table 3 medicina-57-01155-t003:** Perioperative results.

	Era 1 (*n* = 90)	Era 2 (*n* = 87)	Era 3 (*n* = 81)	*p* Value
30-day mortality, *n* (%)	12 (13)	10 (11)	3 (4)	0.07
- Hemorrhagic shock, *n* (%)	5 (5)	4 (5)	1 (1)	
- MOF, *n* (%)	2 (2)	3 (3)	1 (1)	
- Cardiogenic shock, *n* (%)	4 (4)	1 (1)	1 (1)	
- Cerebrovascular accident, *n* (%)	1 (1)	2 (2)	-	
Chest re-exploration, *n* (%)	17 (19)	19 (22)	5 (6)	0.02
Acute renal failure, *n* (%)	38 (42)	34 (39)	41 (51)	0.3
Dialysis, *n* (%)	15 (17)	16 (18)	12 (15)	0.8
High inotropic support, *n* (%)	34 (38)	30 (34)	17 (21)	0.04
Coma, *n* (%)	5 (6)	3 (3)	1 (1)	0.12
- Postoperative onset, *n* (%)	5 (6)	3 (3)	1 (1)	0.12
Neurological focal deficit, *n* (%)	12 (13)	9 (10)	8 (10)	0.23
- Postoperative onset, *n* (%)	9 (10)	5 (6)	3 (4)	0.18
Paraplegia, *n* (%)	1 (1)	1 (1)	2 (2)	0.52
- Postoperative onset, *n* (%)	0 (0)	0 (0)	1 (1)	0.64
Transient ischemic attack, *n* (%)	0 (0)	3 (3)	3 (4)	0.11
Median ICU stay, days (range)	8 (1–26)	9 (1–24)	7 (1–17)	0.25
Median hospital stay, days (range)	20 (5–65)	22 (4–53)	18 (4–49)	0.17

MOF = multiorgan failure; ICU = intensive care unit.

**Table 4 medicina-57-01155-t004:** Univariable and multivariable log-binomial regression analysis of baseline covariates in relation to 30-day mortality.

	Univariable		Multivariable	
	RR (95% CI)	*p* Value	OR (95% CI)	*p* Value
Age	1.01 (0.98–1.04)	0.52		
Male gender	1.42 (0.62–3.29)	0.41		
Chronic renal failure	1.01 (0.40–2.57)	0.98		
Previous cardiac surgery	1.78 (0.47–6.69)	0.39		
Bicuspid aortic valve	2.80 (0.97–8.05)	0.06		
Coronary artery disease	1.90 (0.27–13.30)	0.52		
Previous stroke	5.73 (2.34–14.01)	0.01		
Chronic AF	2.59 (1.13–5.95)	0.02		
Arterial hypertension	2.17 (1.05–4.54)	0.03		
Smoke	1.27 (0.53–3.06)	0.59		
Diabetes	2.57 (0.88–7.49)	0.08		
Tamponade/shock	3.82 (1.76–8.27)	0.001	4.35 (1.88–10.06)	0.001
Syncope	2.36 (0.58–9.67)	0.23		
Neurological damage	2.42 (1.12–5.24)	0.03	2.41 (1.03–5.63)	0.04
Splanchnic malperfusion	1.45 (0.53–3.93)	0.47		
Aortic arch replacement	1.52 (0.59–3.89)	0.38		
AVR mechanical	0.98 (0.25–3.88)	0.98		
AVR biological	3.09 (1.21–7.85)	0.02		
Aortic valve repair	0.67 (0.10–4.66)	0.69		
CABG	2.21 (0.74–6.55)	0.15		
CPB time	1.01 (1.00–1.01)	0.001		
ACC time	1.02 (1.01–1.03)	0.002	1.02 (1.01–1.04)	<0.01
Circulatory arrest	1.01 (1.00–1.01)	0.03		
Antegrade cerebral perfusion	1.46 (0.57–3.74)	0.43		
Retrograde cerebral perfusion	0.46 (0.14–1.48)	0.19		
Era 2 *	0.86 (0.39–1.89)	0.71		
Era 3 *	0.28 (0.08–0.95)	0.04		

AF = atrial fibrillation; AVR = aortic valve replacement; CABG = coronary artery bypass grafting; CPB = cardiopulmonary bypass; ACC = aortic cross-clamp. * Compared to Era 1.

**Table 5 medicina-57-01155-t005:** Late mortality.

	Era 1	Era 2	Era 3
Late deaths, *n*. (%)	41 (45)	18 (21)	5 (6)
Aortic related, *n*. (%)	11 (27)	6 (33)	3 (60)
Cardiac related, *n*. (%)	8 (19)	2 (11)	-
Neoplasia, *n*. (%)	5 (12)	5 (28)	-
Infection, *n*. (%)	10 (24)	2 (11)	2 (40)
Other, *n*. (%)	7 (18)	3 (17)	-

**Table 6 medicina-57-01155-t006:** Reinterventions on thoracic aorta.

*N*. (%)	Era 1 10 (11)	Era 2 12 (14)	Era 3 2 (3)
TEVAR	5(6)	5 (6)	0 (0)
Median time, years (range)	2.2 (0.1–7.1)	0.7 (0.1–1.4)	-
Arch replacement and ET	1 (1)	2 (2)	1 (1)
Arch replacement and FET	2 (2)	0 (0)	0 (0)
Bentall operation	1 (1)	1 (1)	1 (1)
Aortic root remodeling	0 (0)	2 (2)	0 (0)
Proximal pseudoaneurysm repair	1 (1)	1 (2)	0 (0)
Median time, years (range)	3.8 (0.6–6)	1.1 (0.1–2.8)	1 (0.1–1.7)

**Table 7 medicina-57-01155-t007:** Perioperative complications according to the classical and frozen ET.

Perioperative Complications	ET	FET	*p* Value
Chest re-exploration, *n* (%)	2 (8)	0 (0)	0.52
Acute renal failure, *n* (%)	18 (72)	5 (38)	0.07
Dialysis, *n* (%)	8 (30)	0 (0)	0.03
High inotropic support, *n* (%)	9 (34)	2 (15)	0.24
Coma, *n* (%)	1 (4)	0 (0)	0.67
Neurological focal deficit, *n* (%)	3 (12)	1 (8)	0.68
Paraplegia, *n* (%)	2 (8)	0 (0)	0.47
Transient ischemic attack, *n* (%)	5 (20)	1 (8)	0.64
Median ICU stay, days (range)	7 (1–28)	5 (1–24)	0.55
Median hospital stay, days (range)	21 (1–65)	15 (1–53)	0.32

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
