# Peer review of "An Integrated Approach for Treatment of Acute Type A Aortic Dissection"

_medicina, 2021, doi:10.3390/medicina57111155_

Round 1
Reviewer 1 Report
The authors have made appropriate changes
Reviewer 2 Report
The paper is interesting and well writed. I suggest the publication.
This manuscript is a resubmission of an earlier submission. The following is a list of the peer review reports and author responses from that submission.
Round 1
Reviewer 1 Report
The authors descibed their experience in 17 years of surgery of aorta in different time of intervals (3 ERA). They concluded that with increasing experience and a more aggressive approach, including total arch replacement, repair of A-AAD can be performed with low operative mortality in many patients.
Axillary artery cannulation almost routinely used in Era 2 (86%) and 3 (91%) while one femoral artery was mainly cannulated in Era 1 (91%). There was a significant increase of arch replacement procedures from Era 1 (11%) to Era 2 (33%) and 3 (48%). A frozen elephant trunk was mainly performed in Era 3.
The paper is interesting, well writed, good methodology and I suggest minor revision:
Minor revision
1) Add these papers and discuss him in DISCUSSION SECTION:
Off-Label Treatment With Transfemoral Bare Stents for Isolated Aortic Arch Dissection
Di Tommaso, L., Di Tommaso, E., Giordano, R., Pilato, E., Iannelli, G. ,
Treatment with transfemoral bare-metal stent of residual aortic arch dissection after surgical repair of acute type an aortic dissection
Di Tommaso, L., Giordano, R., Di Tommaso, E., Di Palo, G., Iannelli, G. ,
Author Response
We have added the suggested references.
Reviewer 2 Report
While this is a noteworthy topic addressing the various eras of type A aortic dissection, the manuscript needs substantial improvement. Overall, the manuscript lacks statistical and epidemiologic rigor. Below I note a few concerns.
- Clearly, no one proofread this paper before submitting to the journal. There are many formatting and syntax errors, as well as fragmentation issues. For example, in the authorship list, who is “and MD”. In line 25, there needs to be a space after the full stop “repair.Results”. In line 26, the word “was” is missing before “almost”. In line 28, “p” is missing before “<0.01” which is a clear indication that a statistician did not review this manuscript. In line 31, it is not clear what is being compared with respect to the stated p-values. Additionally, the sentence is missing a closing parenthesis. Similar mistakes are found throughout the remaining paper.
- This is basically a retrospective cohort study. As such, log-binomial rather then logistic regression should be used in Table 4 (as risk estimates will be inflated given the common outcome). Also, LR is prone to collapsibility bias. Adjusted Cox-regression should be used for survival analysis.
3) Given the long period of the study, it is difficult to believe that there are no missing values. Missing values can seriously bias study findings and it is important to mention how they were handled in the analysis (e.g., excluded vs. EM algorithm). Additionally, methods and software for data collection likely changed over time and needs to be properly documented.
4) Why were noticeably fewer patients seen in Era 1 (n=90) vs. Era 3 (81). Were certain patients excluded? If so, this raises a serious concern about potential selection bias.
5) Cardiac function parameters such as EF, heart rate, MV E/A, and LAD should be provided. Additionally, a more extensive list of comorbidities/prior medical history is needed (e.g., COPD, MI, HF, PAD,).
6) High inotropic support is vague. Please provide exact numbers and percentages for Dopamine, Epinephrine, Isoproterenol, Milrinone, and Norepinephrine. How many patients received Phosphodiesterase-3 inhibitors, respectively?
7) How many patient (%) had BMI’s ≥30 for each era? Also, please list medications by era (e.g., ACEi/ARBs, Beta blockers, CCBs, Diuretics, DM meds, Nitrates, and Statins).
8) What was the breakdown of cardioplegia by era (e.g., Colloid, Crystalloid, Crystalloid + Oxygen)?
Author Response
While this is a noteworthy topic addressing the various eras of type A aortic dissection, the manuscript needs substantial improvement. Overall, the manuscript lacks statistical and epidemiologic rigor. Below I note a few concerns.
- Clearly, no one proofread this paper before submitting to the journal. There are many formatting and syntax errors, as well as fragmentation issues. For example, in the authorship list, who is “and MD”.
- In line 25, there needs to be a space after the full stop “repair.Results”.
- In line 26, the word “was” is missing before “almost”.
- In line 28, “p” is missing before “<0.01” which is a clear indication that a statistician did not review this manuscript.
Answer: We have verified that some of the errors contained in the printed version are not present in the original submission. However, we have rechecked the entire manuscript and corrected all typing mistakes as all those abovementioned and underlined by the Reviewer.
- In line 31, it is not clear what is being compared with respect to the stated p-values. Additionally, the sentence is missing a closing parenthesis. Similar mistakes are found throughout the remaining paper.
Answer: All the reported p-values come from the comparison between the 3 Groups using one-way ANOVA test or Kruskall–Wallis test for continuous variables, Chi-square or Fisher’s exact test for categorical variables and Log-rank test for survival analysis. According to the reviewer’s comment, we have realized that in several parts of the abstract and the manuscript more than a p value had been erroneously reported and we have corrected accordingly.
- This is basically a retrospective cohort study. As such, log-binomial rather than logistic regression should be used in Table 4 (as risk estimates will be inflated given the common outcome). Also, LR is prone to collapsibility bias. Adjusted Cox-regression should be used for survival analysis.
Answer: According to the reviewer’s suggestion we have performed a univariable and multivariable log-binomial regression analysis to evaluate the relative risk (RR) of baseline covariates associated with 30-days mortality. All potential confounders were initially entered into the multivariable model on the basis of known clinical relevance or imbalances between groups identified in univariable analyses; then a model reduction was performed by excluding variables that did not have any confounding effect, that is, could not make meaningful change (±10%) in the RR (Kleinbaum, et al. 2007. Applied Regression Analysisand Other Multivariable Methods, 4th ed. Belmont, CA).
As suggested by the reviewer we have also performed an adjusted Cox proportional hazards analysis (including clinically relevant confounders: age, gender, LVEF, hypertension, diabetes, chronic kidney disease, coronary artery disease and history of atrial fibrillation) to test the association between the Era of surgery and long-term outcome events. Era 3 patients had a significantly better long-term survival with a 68% lower risk of death compared to Era 1 patients (adjusted HR 0.32, 95% CI 0.14-0.72; p=0.006) while no significant differences were observed for Era 2 patients (adjusted HR 0.80, 95% CI 0.47-1.35; p=0.39). Compared to Era 1 patients, no significant differences in the long-term risk of reintervention were observed among Era 2 patients (adjusted HR 1.54, 95% CI 0.63-3.75; p=0.33) nor Era 3 patients (adjusted HR 0.41, 95% CI 0.08-2.00; p=0.27).
We have added this information to the methods and results of the revised manuscript.
3) Given the long period of the study, it is difficult to believe that there are no missing values. Missing values can seriously bias study findings and it is important to mention how they were handled in the analysis (e.g., excluded vs. EM algorithm). Additionally, methods and software for data collection likely changed over time and needs to be properly documented.
Answer: All the data that we have considered important for the aim of the paper were available and not missing in our database and reported in the paper.
4) Why were noticeably fewer patients seen in Era 1 (n=90) vs. Era 3 (81). Were certain patients excluded? If so, this raises a serious concern about potential selection bias.
Answer: No patient was excluded. In Era 3 there are less patients because Era 3 has a shorter duration (see Methods Line 71).
5) Cardiac function parameters such as EF, heart rate, MV E/A, and LAD should be provided. Additionally, a more extensive list of comorbidities/prior medical history is needed (e.g., COPD, MI, HF, PAD,).
Answer: According to the reviewer’s suggestion we have implemented Table 1 with a more complete list of baseline patient characteristics.
6) High inotropic support is vague. Please provide exact numbers and percentages for Dopamine, Epinephrine, Isoproterenol, Milrinone, and Norepinephrine. How many patients received Phosphodiesterase-3 inhibitors, respectively?
Answer: As high inotropic support we mean high dosages of a single or combination of inotropes. We have used almost exclusively dopamine, epinephrine and norepinephrine as felt indicated but in our database this data is not considered.
7) How many patient (%) had BMI’s ≥30 for each era? Also, please list medications by era (e.g., ACEi/ARBs, Beta blockers, CCBs, Diuretics, DM meds, Nitrates, and Statins).
Answer: These data are not available and we believe not relevant.
8) What was the breakdown of cardioplegia by era (e.g., Colloid, Crystalloid, Crystalloid + Oxygen)?
Answer: In our center only the St. Thomas 1 cardioplegia has been used. This has been added to the Methods (Lines 108,109).
Reviewer 3 Report
Authors present an observational retrospective study of 258 patients that underwent surgical repair of Acute Aortic Dissection type A, observed and operated at a single institution during the time-span of 17 years at the Cardio-thoracic Unit of Udine, a city of north-eastern Italy. The 17 years time interval was divided into 3 phases, during which important changes in prompt diagnosis, organization of the Unit, technical progress and patient care and experience of surgeons were progressively increased. Results show an impressive improvement of in-hospital and delayed mortality, and decrease of complications.
The study is interesting and supplies important notions and data for surgical treatment of this acute, life-treatening condition.
Author Response
We thank the Reviewer for his comments and consideration given to our paper.